

# Identification, expression, and phylogenetic analyses of terpenoid biosynthesis-related genes in secondary xylem of loblolly pine (*Pinus taeda* L.) based on transcriptome analyses

Jipeng Mao[1], Zidi He[1], Jing Hao[1], Tianyi Liu[1], Jiehu Chen[2] and Shaowei Huang[1]

[1] College of Forestry and Landscape Architecture, South China Agricultural University, Guangzhou, Guangdong, China
[2] Science Corporation of Gene, Guangzhou, Guangdong, China

## ABSTRACT

Loblolly pine (*Pinus taeda* L.) is one of the most important species for oleoresin (a mixture of terpenoids) in South China. The high oleoresin content of loblolly pine is associated with resistance to bark beetles and other economic benefits. In this study, we conducted transcriptome analyses of loblolly pine secondary xylem to gain insight into the genes involved in terpenoid biosynthesis. A total of 372 unigenes were identified as being critical for oleoresin production, including genes for ATP-binding cassette (ABC) transporters, the cytochrome P450 (CYP) protein family, and terpenoid backbone biosynthesis enzymes. Six key genes involved in terpenoid biosynthetic pathways were selected for multiple sequence alignment, conserved motif prediction, and phylogenetic and expression profile analyses. The protein sequences of all six genes exhibited a higher degree of sequence conservation, and upstream genes were relatively more conserved than downstream genes in terpenoid biosynthetic pathways. The N-terminal regions of these sequences were less conserved than the C-terminal ends, as the N-terminals were quite diverse in both length and composition. The phylogenetic analyses revealed that most genes originated from gene duplication after species divergence, and partial genes exhibited incomplete lineage sorting. In addition, the expression profile analyses showed that all six genes exhibited high expression levels during the high-oleoresin-yielding phase.

# INTRODUCTION

The loblolly pine's amenability to plantation management, high wood yields, and fast growth make it one of the most economically important forest species in the world (*Lu et al., 2017*). Loblolly pine is also a potential renewable feedstock for alternative energy and fuel, as well as an important source for oleoresin (*Olatunde et al., 2017*). Oleoresin is a viscous mixture of terpenoids stored in resin canals or blisters in the stems of conifers (*Trapp*

Corresponding author
Shaowei Huang,
shwhuang@scau.edu.cn

& *Croteau, 2001*). Resin canals are a characteristic structure of *Pinus*, and as transport channels for resin, they are closely related to pine resistance to pests. The resin canals distributed in secondary vascular structures can be divided into transverse and vertical resin canals. In the secondary xylem in particular, the transverse and vertical resin canals form a two-dimensional network structure; this two-dimensional resin canal structure allows the pine plant to effectively secrete resin at any point of invasion when subjected to biotic stresses (*Chen & Yuan, 2002*). In conifers, the yield of oleoresin is a quantitative trait under moderate genetic control. Meanwhile, the production of oleoresin is correlated with growth traits and meteorological factors (*An & Ding, 2010*). Oleoresin plays a key role in the defense system of coniferous trees, is widely used for industrial chemicals, and can be explored as a precursor for biofuels (*Yamada et al., 2015*). However, partial volatile terpenoids are also inducible upon insect herbivory, oviposition, or fungal inoculation. These relationships are ancient and complex (*Scott, Anderson & Anderson, 2004*).

Terpenoids, an important secondary metabolite in plants, are derived from the five-carbon biosynthetic building blocks isopentenyl diphosphate (IPP) and its isomer, dimethylallyl diphosphate (DMAPP). IPP and DMAPP are formed in two independent pathways: the mevalonate (MVA) and 2-C-methylerythritol-4-phosphate (MEP) pathways, localized in the cytosol/endoplasmic reticulum and plastids, respectively (*Eisenreich et al., 1998*; *Rohmer, 1999*). Condensation of IPP and DMAPP is catalyzed by prenyltransferases and yields three central intermediates of the isoprenoid pathway: geranyl diphosphate (GPP), farnesyl diphosphate (FPP), and geranylgeranyl diphosphate (GGPP). The numerous structurally diverse plant terpenoids are then formed from these prenyl diphosphate precursors by terpene synthases (*Miller, Oschinski & Zimmer, 2001*). During the biosynthesis of terpenoids, there are many other terpenoid backbone biosynthesis key enzymes involved, such as 1-deoxy-D-xylulose 5-phosphate synthase (DXS) and 1-deoxy-D-xylulose 5-phosphate reductoisomerase (DXR) are the first two step enzymes of MEP pathway and play a critical role in the overall regulation of the pathway (*Lois, Gallego & Campos, 2000*; *Cordoba, Salmi & León, 2009*). 3-hydroxy-3-methylglutaryl-CoA reductase (HMGR) is a key enzyme of the MVA pathway (*Rodríguez-Concepion & Boronat, 2015*). Isopentenyl diphosphate isomerase (IPPI) acts on isomerization between IPP and DMAPP, which are the precursors to all isoprenoid compounds (*Zulak & Bohlmanu, 2010*). The geranyl diphosphate synthase (GPS), geranylgeranyl diphosphate synthase (GGPS) and farnesyl diphosphate synthase (FPS) are the precursors to monoterpenes, diterpenes and sesquiterpenes of conifer oleoresin, respectively (*Zulak & Bohlmanu, 2010*). Furthermore, previous studies have confirmed that ABC transporters (*Jasinski et al., 2001*), several cytochrome P450 family proteins (*Banerjee & Hamberger, 2018*), aldehyde/alcohol dehydrogenases (ALDH/ADH) (*Teoh, Polichuk & Reed, 2009*; *Polichuk et al., 2010*), pathogenesis-related proteins (*Dafoe et al., 2009*), non-specific lipid-transfer proteins (*Liu et al., 2015*), and ethylene responsive transcriptionfactors (*ERFs*) (*Xu et al., 2007*) are also crucial for terpenoid biosynthesis.

With the decreasing cost of sequencing, transcriptome analyses based on RNA-Seq technology has been widely used in many plant species to identify target genes or acquire other useful genomic information. For example, transcriptome analyses of *Pinus*

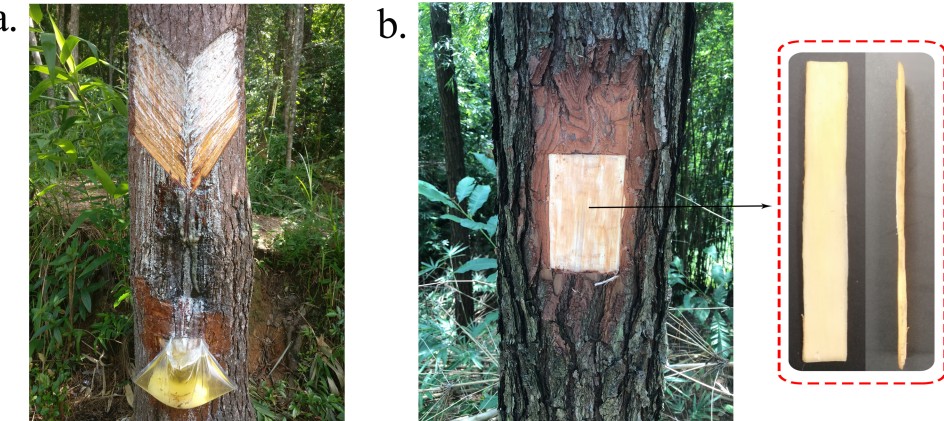

**Figure 1** **Oleoresin and RNA samples collection methods.** (A) Artificial collection of oleoresin in South China. (B) Sampling site and method of secondary xylem tissue.

*massoniana* (*Liu et al., 2015*), *Lindera glauca* (*Niu et al., 2015*), *Thapsia laciniata* (*Drew et al., 2013*), *Isodon rubescens* (*Su et al., 2016*), and *Calotropis procera* (*Pandey et al., 2016*) were conducted to identify candidate genes related to terpenoid biosynthesis. The loblolly pine (*Pinus taeda*) genome has been sequenced (*Zimin et al., 2014*) and contains 20.15 billion bases, approximately seven times that of the human genome. It is the largest sequenced genome and the most complete published conifer genome sequence, yet few studies have reported transcriptome analyses of *Pinus taeda* secondary xylem. We performed transcriptome analyses of loblolly pine secondary xylem for two reasons: first, the secondary xylem is the main tissue involved in the secretion of terpenoids; second, the collection of resin in South China is based on the bark streak method (Fig. 1A) instead of using the pine needles as the material for alcohol extraction. We conducted an integrated analyses of terpenoid biosynthesis encompassing the identification of related genes, the phylogenetic relationships among representative genes in other characterized species, the expression profiles of different oleoresin-yielding stages, and the prediction of conserved motifs. We provide an extensive perspective of regulatory factors involved in terpenoid biosynthesis in the loblolly pine, as well as a portfolio of candidate genes for future study.

## MATERIALS AND METHODS

### Plant materials

Because of the content of oleoresin was different between individuals, and influenced by the meteorological factors. In this study, three 15-year-old loblolly pine trees from the Yingde Research Institute of Forestry in Guangdong Province, China, were used to RNA sequencing to find out the genes involved in terpenoid biosynthetic pathway. They shared no lineage relationships. During the high-oleoresin-yield stage (August) (*An & Ding, 2010*), secondary xylem tissue samples ∼1 mm in thickness were collected from the trees after removing the bark at breast height using a firewood chopper and removing the phloem

and cambium using a 75% alcohol-treated grafting knife (Fig. 1B). The samples were immediately immersed in liquid nitrogen and then stored at −80 °C until RNA extraction.

## RNA isolation

Total RNA was extracted from each sample following a protocol modified from *Chang, Puryear & Caimey (1993)*. Briefly, secondary xylem tissues (∼150 mg) were ground into powder under liquid nitrogen, then immediately transferred to 1.5 mL RNase-free centrifuge tubes. Next, 1 mL CTAB extraction buffer (100 mM Tris-HCl, pH 8; 25 mM EDTA, pH 8; 2 M NaCl; 2% CTAB; 2% PVP) and 20 μL β-mercaptoethanol were added to the tubes, mixed by vortexing for 2 min, and incubated at 65 °C for 20 min. Following incubation, samples were centrifuged for 10 min at 12,000 rpm at 4 °C. The supernatant was collected into a new 1.5 mL centrifuge tube and an equal volume of chloroform/isoamyl alcohol (24:1) was added, mixed by vortexing for 2 min, and centrifuged for 5 min at 12,000 rpm at 4 °C. After an additional chloroform/isoamyl alcohol extraction, the supernatant was collected into a new tube and a $\frac{1}{4}$-volume of 10 M LiCl was added, mixed, and incubated at 4 °C for 6 h. After precipitation for 6 h, the sample was centrifuged for 10 min at 12,000 rpm at 4 °C, and the supernatant was discarded. The sediment was washed twice with 500 μL 75% EtOH and 500 μL anhydrous alcohol precooled at −20 °C, respectively. The sample was dried for 5 min, then resuspended in 50 μL DEPC-treated water. Total RNA concentration and integrity were determined, and high-quality RNAs were used for cDNA library construction.

## cDNA library construction, sequencing, and sequence assembly

cDNA libraries were constructed following the protocol described by *Foucart et al. (2006)*. Libraries were sequenced with $2 \times 100$ paired-end reads using the Illumina HiSeq 4000 platform at Science Corporation of Gene (Guangzhou, China). The Trinity method (*Haas et al., 2013*) was used to assemble the high-quality reads into unigenes. RNA sequence data were deposited in the National Center for Biotechnology Information (NCBI) Sequence Read Archive (SRA) under the study accession number PRJNA482703.

## Functional annotation and identification of terpenoid biosynthesis-related genes

To identify genes related to terpenoid biosynthesis, all unisequences were queried against five public protein databases (UniProt, Nr, KEGG, KOG, and GO) with an *E*-value threshold $<10^{-5}$ to detect unigenes homologous to previously identified relevant genes. The Blast2GO software (version 4.1.5) with the default parameters (*Conesa, Terol & Robles, 2005*) was used to obtain GO annotation of unigenes. WEGO (*Ye et al., 2006*) was used to perform GO functional categorization for all unisequences. KEGG pathway analyses were performed using BLASTx queries against the KEGG database (http://www.genome.jp/kegg/).

## Multiple sequence alignment analyses and conserved motif prediction

Multiple sequence alignments were performed using ClustalW (https://www.genome.jp/tools-bin/clustalw) and images were displayed using a multi-sequence comparison

display tool (http://www.bio-soft.net/sms/multi_align.html). The conserved motif predictions were performed using the Multiple Em for Motif Elicitation (MEME) tool (http://meme.sdsc.edu/meme/cgi-bin/meme.cgi) (*Bailey & Elkan, 1994*) with the default parameters.

## Phylogenetic analyses

Phylogenetic analyses were performed based on the protein sequences of six completely assembled representative genes (if more than one homologous unigene was fully assembled, the unigene with the largest sequence length was used for subsequent analyses); alignments were performed using the MUSCLE (v3.8.30) program. Phylogenetic trees were constructed using two methods: maximum likelihood (ML) in RAxML (8.1.5) with 1,000 bootstrap replicates in view of the best-scoring model, and Bayesian inference (BI) in MrBayes (3.2.6) with 10,000,000 generations. The best models were selected using ProtTest3, and the BI tree was reliable with a standard deviation of split frequencies <0.01. The phylogenetic trees were drawn with MEGA 7.0 (*Kumar, Stecher & Tamura, 2016*).

## Quantitative reverse transcription PCR (qRT-PCR) analyses

To investigate the expression profiles of six representative genes involved in the biosynthesis of terpenoids, total RNAs from three different oleoresin-yielding stages were extracted as described for the cDNA library preparation. Samples from two low-oleoresin-yield phases were collected in April and October; the high-oleoresin-yield RNA samples (August) were the same samples used for cDNA library construction. Reverse transcription was performed using a PrimeScript $^{TM}$ RT reagent kit with a gDNA Eraser (Code No: RR047A; TaKaRa, Dalian, China) following the manufacturer's instructions. The primers for the six candidate gene sequences were designed using Primer Premier 5.0, and *Actin* (F: gagcaaagagatcactgcacttg; R: ctcatattcggtcttggcaatcc) was selected as an internal control (primers are listed in The qRT-PCR analyses were performed using a LightCycler 480 System (Roche, Basel, Switzerland) and a TB Green *Premix Ex Taq* II kit (Code No: RR820A; TaKaRa). A 20 μL reaction containing 2 μL synthesized cDNA, 10 μL 2×TB Green *Premix Ex Taq* mix, 0.8 μL 10 μM forward primer, 0.8 μL 10 μM reverse primer, and 6.410 μL sterile distilled water was amplified using the following procedure: one cycle of 95 °C for 30 s, followed by 40 cycles of 95 °C for 5 s, 55 °C for 30 s, and 72 °C for 30 s. Three biological replicates (consistent with the samples used for library construction) of each oleoresin-yielding stage were performed, and each PCR reaction included three technical replicates. The relative expression levels of candidate genes were calculated using the $2^{-\triangle\triangle Ct}$ method (*Livak & Schmittgen, 2001*).

# RESULTS

## RNA-Seq and transcriptome assembly

To obtain a summarization of the loblolly pine transcriptome in secondary xylem tissues during the high-oleoresin-yielding phase (August), total RNAs extracted from three samples were used for RNA-Seq on an Illumina HiSeq 4,000 platform. After quality appraisal and low-quality data screening, 29,559,842 high-quality reads were generated and assembled

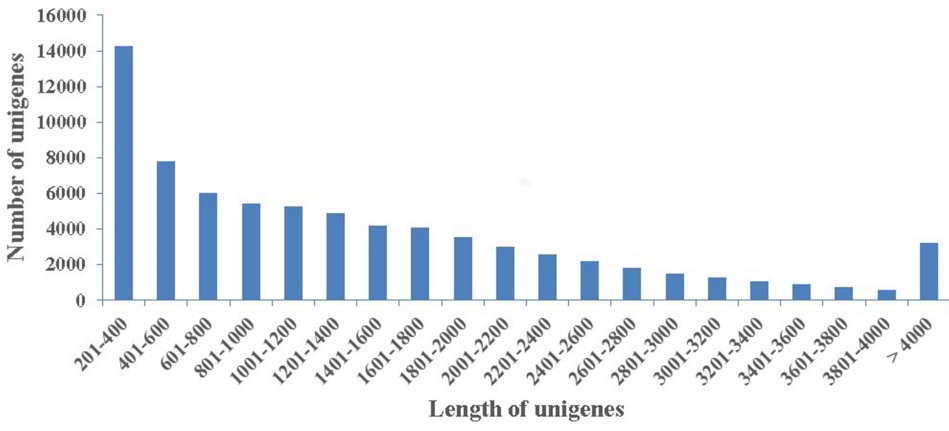

**Figure 2** **Length distribution.** The length distribution of 74,402 assembled unigenes in secondary xylem of *Pinus taeda*.

**Table 1** **Summary of unigenes functional annotation.** List of 74,402 unigenes against five public database.

| Database | Number of annotated unigenes | Percentage of annotated unigenes (%) |
|---|---|---|
| Gene Ontology (GO) | 22,433 | 30.15% |
| euKaryotic Orthologous Groups (KOG) | 24,871 | 33.43% |
| Kyoto Encyclopedia of Genes and Genomes (KEGG) | 27,942 | 37.56% |
| Non-redundant (Nr) | 30,395 | 40.85% |
| UniProt | 31,370 | 42.16% |

into 115,399 transcripts with a mean length of 688 bp and an N50 of 1,255 bp. Among these transcripts, 20,757 (17.98%) were >1 kb. Finally, we obtained 74,402 unigenes with a mean length of 1,459 bp (Fig. 2).

## Functional annotation and classification

A total of 31,586 unigenes (42.45%) were annotated with a threshold $E$-value $<10^{-5}$ after BLASTx searches against the five protein databases (Table S2). Among them, 31,370 (42.16%), 30,395 (40.85%), 27,942 (37.56%), 24,871 (33.43%), and 22,433 (30.15%) unigenes were annotated by the UniProt, Nr, KEGG, KOG, and GO databases, respectively (Table 1, Fig. 3). Based on the Nr database, 5,807 (18.38%) unigenes exhibited significant homology with sequences from *Picea sitchensis*, and 1,877 (6.10%) and 1,305 (4.13%) unigenes shared high similarity with sequences from *Amborella trichopoda* and *Nelumbo nucifera*, respectively (Fig. S1).

GO assignments based on sequence homology were used to classify the functions of annotated unigenes. In all, 22,433 unigenes were distributed into three main GO terms ("biological process", "cellular component" and "molecular function") and 44 secondary-class GO terms (Fig. 4). The two largest GO functional groups of the "molecular function" term were "catalytic activity" and "binding". The most represented "cellular component"

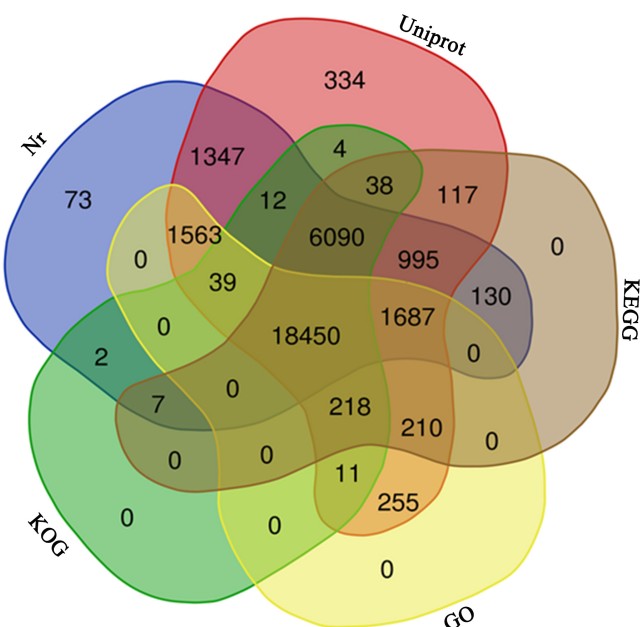

**Figure 3  Venn diagram.** The venn diagram of 31,586 unigenes from five public database.

categories were "membrane part" and "cell part". The most represented "biological process" categories were "metabolic process", "single organism process" and "cellular process". Several important secondary-class GO terms related to terpenoid metabolism were also represented at lower numbers, such as "response to stimulus", "transporter activity" and "biological regulation" (Fig. 4). In general, the GO analyses revealed that most unigenes were responsible for basic biological process regulation and metabolism.

To further enhance the reliability of the transcriptome annotation process, the KOG database was used to classify the functional groups of all annotated unigenes. In our study, 18,786 matched unigenes were classified into 25 groups (Fig. 5). Among these groups, the largest number of unigenes (7419, 39.49%) were classified as "general function prediction only". In addition, some groups related to the metabolism of terpenoids such as: "lipid transport and metabolism" (549 unigenes), "secondary metabolites biosynthesis, transport, and catabolism" (674 unigenes), and "defense mechanisms" (148 unigenes) were also highly represented. However, only 50, 38, and three unigenes matched "nuclear structure", "extracellular structure" and "motility", respectively. Functional classification of unigenes focusing on biochemical pathways was conducted using the KEGG annotation system. In our study, 10,266 unigenes were assigned to 35 pathways (Fig. 6). Most plant biochemical pathways were represented, including "metabolism", "organismal systems", "genetic information processing", "cellular processes" and "environmental information processing". The most represented pathway was "global and overview maps" (2031, 19.78%). In addition, a total of 123 unigenes were involved in the terpenoid biosynthetic pathway (Figs. S2–S4, Table S3).
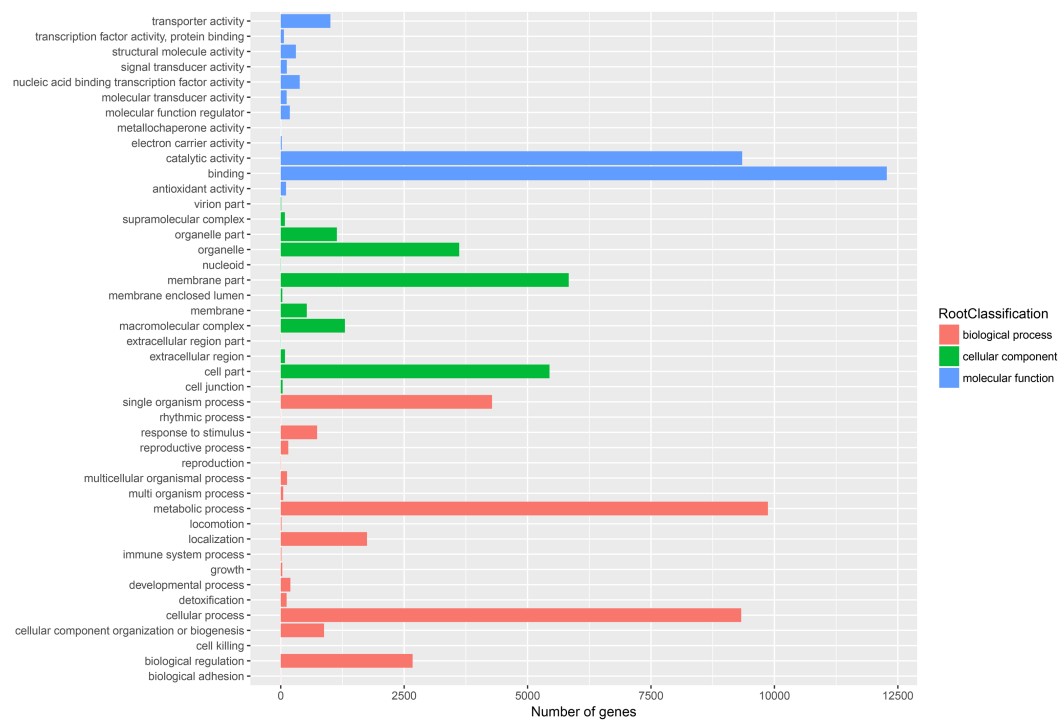

**Figure 4** **Assembled unigenes were functionally classified by Gene Ontology categorization.** The unigenes corresponded to three main categories: biological process, cellular component, and molecular function.

## Identification of oleoresin biosynthesis-related genes

After combining the functional annotation with previous research results, a total of 249 unigenes (not including the 123 unigenes involved in the terpenoid biosynthetic pathway) were identified as crucial for oleoresin biosynthesis. Among these, 105 ABC transporter genes were identified and mainly divided into seven subfamilies (A, B, C, D, F, G, and I) (Table S4). In all, 67 genes were annotated as cytochrome P450 family proteins, including many known subfamilies related to terpenoid biosynthesis, such as cytochrome P450 family members CYP720, CYP71, CYP701, CYP51, CYP76, CYP704, and CYP716 (Table S5). In addition, there were 20 alcohol dehydrogenase genes, 10 aldehyde dehydrogenase genes, 17 pathogenesis-related protein (PR5 and PR10) genes, 13 ethylene-responsive transcription factor genes, 13 terpenoid synthase genes (Table 2), three non-specific lipid-transfer protein genes, and one phosphomethylpyrimidine synthase gene (Table S6).

## Multiple sequence alignment and conserved motif prediction

BLASTP searches against the Nr database for comparative sequence analyses of DXR, DXS, HMGR, IPPI, GGPS, and FPS proteins were conducted separately. A total of 14 additional representative plant species containing the six protein sequences were retrieved (if multiple protein sequences were retrieved for a species, the one with the highest BLASTP score was selected for analyses) (Table 3). Among them were three gymnosperm species, four monocotyledons, and seven dicotyledons. Multiple sequence alignment analyses revealed

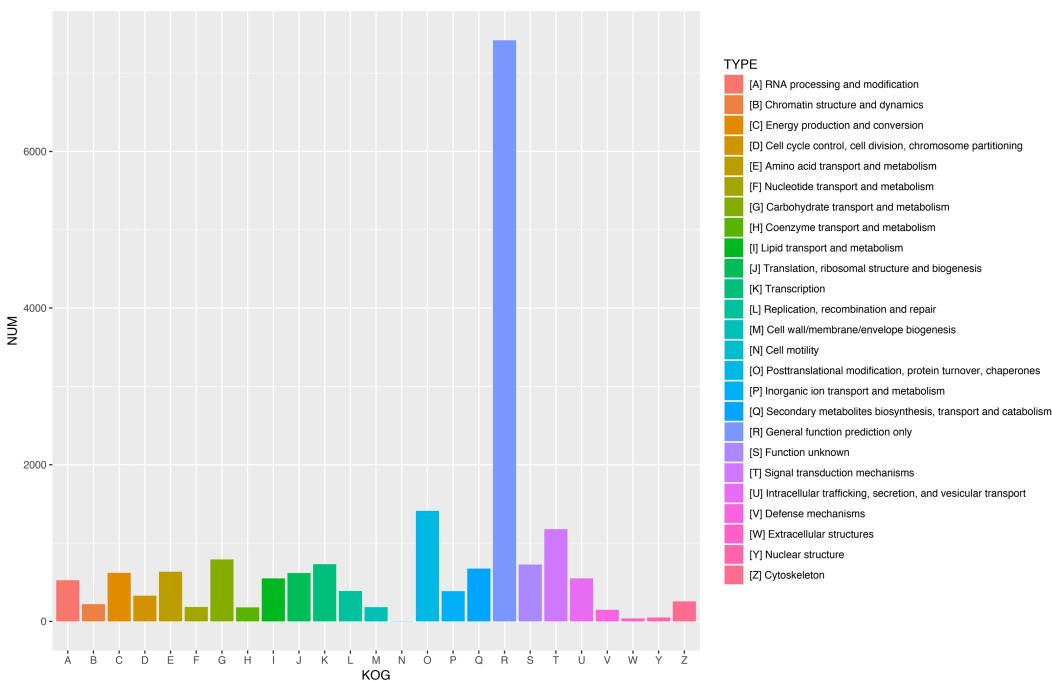

**Figure 5  KOG functional classification of all unigenes.** A total of 18,786 unigenes showed significant similarity to the sequences in KOG databases and were clustered into 25 categories.

that the FPS (Fig. S5) and GGPS (Fig. S5F) sequences were less conserved compared to the other four protein sequences, and the C-terminal regions of the six protein sequences were more conserved than the N-terminal ends. The N-terminals of these protein sequences were quite diverse in both length and composition (Fig. S5).

In our study, the MEME motif search tool was used to discover the motifs of the 15 DXR, DXS, HMGR, IPPI, GGPS, and FPS protein sequences, respectively. The results identified a total of 10 motifs for each sequence (except the IPPI protein sequence, which had 9), named motifs 1–10, separately (Fig. 7 and Fig. S6). For the DXR protein sequences, six identical instances of motif 3, and two identical instances of motifs 7, 8, and 9 were discovered in different locations, respectively. All members contained motifs 2, 3a, 4, 3b, 5, 3c, 6, 7b, 8a, 9b, and 10. Interestingly, the gymnosperm clade were all missing motifs 5, 3d, and 8b (except the *Picea sitchensis* DXR sequence) (Fig. 7). Among the DXS protein sequences, motifs 5–10 were more conserved than other motifs. Excepting the *Taxus × media* DXS sequence, the other 14 members contained motif 1 (Fig. S6A). Analyses of conserved sequence motifs of IPPI sequences revealed that motifs 4–8 were present in all members, and all but *Taxus × media* contained motif 9. Motifs 1–3 were apparently less conserved as they were only detected in two, six, and one members, respectively (Fig. S6C). For the HMGR protein sequences, 13 members contained motifs 1–10; the *Elaeis guineensis* HMGR sequence lost motif 10 and *Pinus taeda* was missing motifs 6–10 (Fig. S6B). For FPS protein sequences, motifs 2–5 and 7–10 were present in all 15 members; only *Oryza sativa japonica* was missing motif 1. Interestingly, four of the five members containing

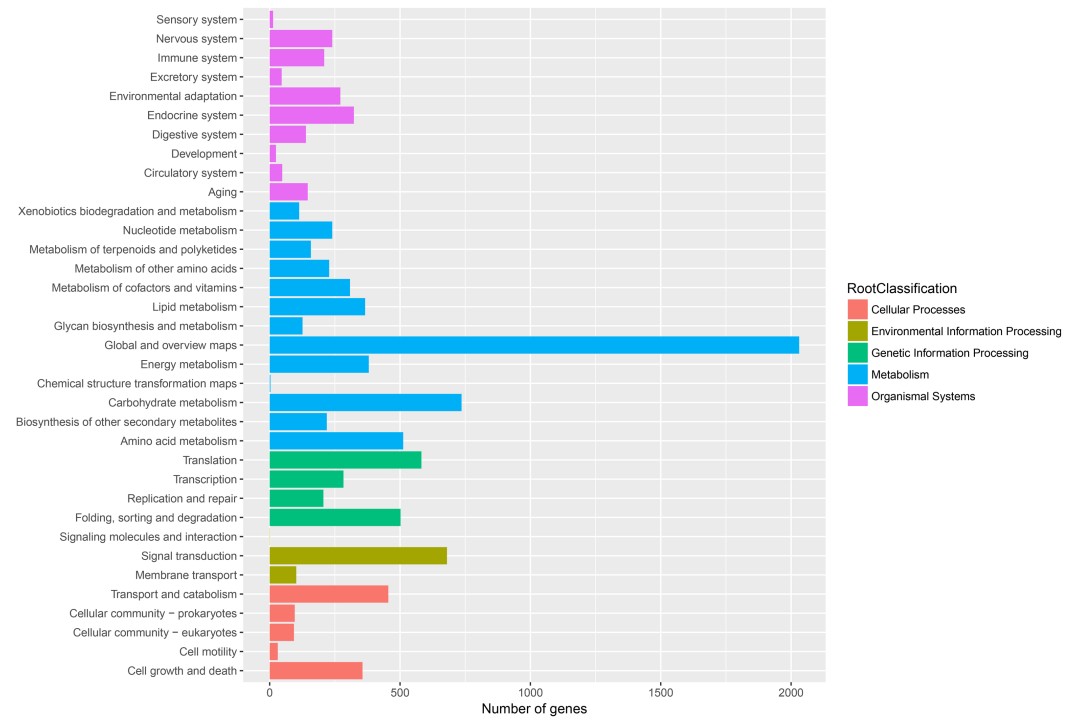

**Figure 6  Assembled unigenes were functionally classified by KEGG classification.** The unigenes corresponded to five main categories: Cellular processes, Environmental information Processing, Genetic information Processing, Metabolism and Organismal Systems.

**Table 2  List of 13 terpenoid synthase unigenes.** Summary of 13 terpenoid synthase unigenes and their functional annotation information.

| Classification | Unigene_ID | Unigene length (bp) | Nr_annotation | Accession number |
|---|---|---|---|---|
| Monoterpenoid | unigene065907_i1 | 387 (partial) | Carene synthase | Q84SM8.1 |
| | unigene056578_i1 | 2,238 (partial) | (-)-limonene synthase | AAS47694.1 |
| | unigene003908_i1 | 2,265 (complete) | (+)-alpha-pinene synthase | Q84KL3.1 |
| | unigene108418_i10 | 3,432 (complete) | farnesene synthase | ADH29869.1 |
| Diterpenoid | unigene044752_i1 | 251 (partial) | alpha-humulene synthase | ADZ45513.1 |
| | unigene108420_i1 | 360 (partial) | alpha-bisabolene synthase | AAS47689.1 |
| Sesquiterpenoid | unigene057627_i12 | 1,206 (complete) | levopimaradiene synthase | M4HXU6.1 |
| | unigene011223_i2 | 1,309 (complete) | momilactone a synthase | OIT37089.1 |
| Triterpenoid | unigene087027_i1 | 2,055 (complete) | squalene synthase | AHI96421.1 |
| | unigene037971_i1 | 3,162 (complete) | cycloartenol synthase | AAG44096.1 |
| Tetraterpenoid | unigene082221_i1 | 782 (partial) | Trans-phytoene synthase | ABR16280.1 |
| | unigene082222_i1 | 714 (partial) | Cis-phytoen synthase | ABR16198.1 |

motif 6 came from the gymnosperm clade (Fig. S6E). For GGPS protein sequences, only four motifs were present in all 15 members, and only the three gymnosperm members contained motif 2 (Fig. S6D).

**Table 3  The protein sequences information of the six representative genes in 14 plant species.** Nr_ID: accession number in Nr dabtabase, Chr: chromosome location, '.' represented unknown chromosome location.

| Species | Gene | | | | | | | | | | | |
|---|---|---|---|---|---|---|---|---|---|---|---|---|
| | DXR | | DXS | | HMGR | | IPPI | | GGPS | | FPS | |
| | Nr_ID | Chr | Nr_ID | Chr | Nr_ID | Chr | Nr_ID | Chr | Nr_ID | Chr | Nr_ID | Chr |
| *Picea sitchensis* | ACN40563.1 | . | ACN39837.1 | . | ACN40476.1 | . | ACN41037.1 | . | ACN40956.1 | . | ACN40171.1 | . |
| *Ginkgo biloba* | AAR95700.1 | . | AAR95699.1 | . | AAU89123.1 | . | ACU56979.1 | . | AAQ72786.1 | . | AAR27053.1 | . |
| *Taxus ×media* | AAU87836.1 | . | AAS89342.1 | . | AAQ82685.1 | . | ALA48967.1 | . | AFD32422.1 | . | AAS19931.1 | . |
| *Zea mays* | ONM30686.1 | 1 | ABP88135.1 | 5 | XP_008677153.1 | 4 | XP_008649644.1 | 8 | NP_001183930.1 | 2 | NP_001105039.1 | 8 |
| *Elaeis guineensis* | XP_010940050.1 | 3 | NP_001290502.1 | 3 | XP_019707817.1 | 8 | XP_010943604.1 | 6 | XP_010905229.1 | . | XP_010924109.1 | 6 |
| *Oryza sativa Japonica* | XP_015618768.1 | 15 | XP_015640505.1 | 14 | XP_015648250.1 | 8 | AAT94033.1 | 1 | XP_015626863.1 | 2 | XP_015628129.1 | 1 |
| *Ananas comosus* | XP_020097704.1 | . | XP_020096323.1 | . | XP_020090296.1 | . | XP_020106556.1 | . | XP_020097093.1 | . | XP_020083542.1 | . |
| *Populus trichocarpa* | XP_002305413.1 | 4 | XP_006380580.2 | 7 | XP_002301898.2 | 2 | XP_002325469.2 | . | XP_002305196.3 | 4 | XP_002308751.2 | 6 |
| *Glycine max* | NP_001240062.1 | 5 | XP_003533424.1 | 9 | XP_003519474.1 | 2 | XP_003534511.1 | 9 | XP_003547995.1 | 16 | XP_003534984.1 | 9 |
| *Eucalyptus grandis* | XP_010027708.1 | . | XP_010024490.1 | . | XP_010061229.1 | . | XP_010051591.1 | . | XP_010062252.1 | . | XP_010057769.1 | . |
| *Nicotiana attenuata* | XP_019261019.1 | . | XP_019253532.1 | 9 | XP_019234273.1 | . | XP_019245686.1 | 8 | ABQ53935.1 | . | XP_019251827.1 | 8 |
| *Vitis vinifera* | XP_002282761.1 | 17 | XP_002277919.1 | 5 | XP_002275827.1 | 18 | NP_001304059.1 | 19 | XP_002283364.1 | 18 | NP_001267864.1 | 19 |
| *Morus notabilis* | XP_010101212.1 | . | EXC03145.1 | . | XP_010094509.1 | . | XP_010111085.2 | . | XP_010095287.1 | . | XP_010103952.1 | |
| *Arabidopsis thaliana* | NP_201085.1 | 5 | NP_193291.1 | 5 | NP_177775.2 | 1 | NP_197148.3 | 5 | NP_195399.1 | 4 | NP_199588.1 | 5 |

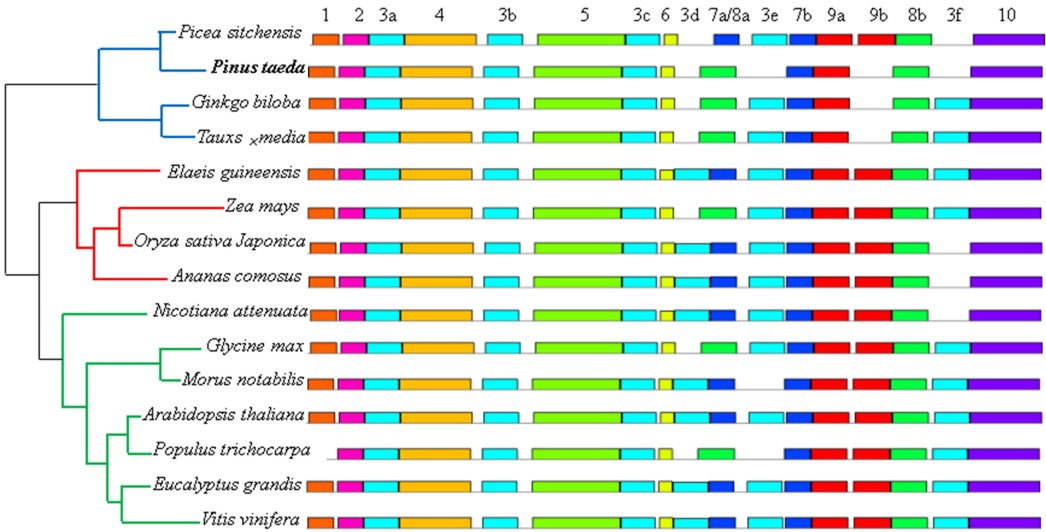

**Figure 7** **Distribution of conserved motifs of DXR protein sequences in 15 plant species.** Phylogenetic relationship were displayed on left, and blue, red and green lines represented gymnosperm, monocotyledon and dicotyledon, respectively. Each motif is represented by a colored box and a number, and the same number with differ letters represented the identical motif located in different sites.

## Phylogenetic analyses

To further investigate the evolutionary relationships of *Pinus taeda* terpenoid backbone biosynthesis-related genes to those from other plants, six genes annotated as encoding terpenoid backbone biosynthesis key enzymes were selected. Among them were three upstream genes (*DXS*, *DXR*, and *HMGR*), two downstream genes (*FPS* and *GGPS*), and one gene located at the branching point (*IPPI*) (Fig. 8). For each gene, a phylogenetic tree was built using amino acid sequences from 15 species (Fig. 9). Phylogenetic analyses results showed that *DXS*, *DXR*, *HMGR*, and *IPPI* had the same unrooted topology. Amino acid sequences of *DXS*, *DXR*, *HMGR*, and *IPPI* genes from the 15 species were divided into three distinct monophyletic clades representing gymnosperms, dicotyledons, and monocotyledons (Figs. 9A, 9B, 9E, 9F). According to the branch lengths of each species in the phylogenetic tree, we speculate that the degree of evolution of these four genes roughly corresponds to the species divergence and traditional classification. However, *FPS* and *GGPS* exhibited anomalous unrooted molecular trees. The four *FPS* genes from the monocotyledons were divided into two branches, and the *FPS* gene of *Ananas comosus* had the longest branch length (Fig. 9D). In the phylogenetic tree of *GGPS* genes, the branch containing the two monocotyledons *Oryza sativa japonica* and *Elaeis guineensi* s displayed a significantly higher degree of evolution than the other branches. In addition, the *Zea mays* (a typical monocotyledon) *GGPS* gene was clustered into the dicotyledon clade (Fig. 9C).

Based on the branching patterns of the phylogenetic trees, roughly similar tree topologies were seen for all genes in the gymnosperm and monocotyledon clades, respectively. For *Pinus taeda* in the gymnosperm clade, except for the *DXS* gene (orthologous to the *Ginkgo biloba* gene), the other five genes were orthologous to the *Picea sitchensis* genes (Fig. 9).

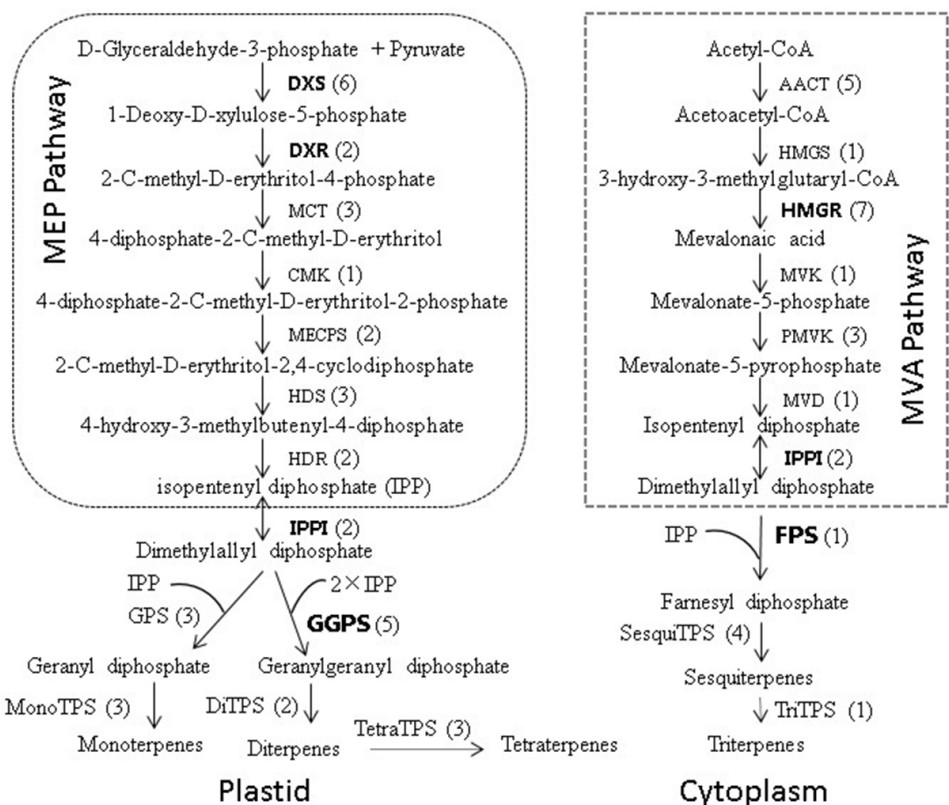

**Figure 8  Biosynthetic pathway of terpenoids (adapted from *Zulak & Bohlmanu (2010)*, *Liu et al. (2015)*).** The number of unigenes homologous to gene families encoding these enzymes was provided in parentheses.

A similar situation also occurred in monocotyledon clade. For *Oryza sativa japonica*, excepting the *GGPS* gene (orthologous to the *Elaeis guineensis* gene), the remaining five genes were orthologous to the *Zea mays* genes. However, in the dicotyledon clade, *Glycine max* and *Morus notabilis* were clustered on the same branches in the *DXR* and *DXS* gene trees, while they were on different branches in phylogenetic trees of the other four genes. The *DXR* and *HMGR* genes from *Populus trichocarpa* and *Arabidopsis thaliana* were more closely related than the *DXR* and *HMGR* genes of the other dicotyledons. This implies that many gene duplication events have occurred independently in different species, regardless of their true species trees. Still, the *DXS* and *IPPI* genes of *Arabidopsis thaliana*, a representative dicotyledon, branched separately from the other dicotyledon sequences.

## Expression profile analyses by qRT-PCR

To further understand the expression profiles of the six representative genes, RNA samples from three different oleoresin-yielding stages were subjected to qRT-PCR analyses. All six were up-regulated in August compared to April and October (Fig. 10). Compared to expression levels in April, four genes (*IPPI*, *DXS*, *FPS*, and *DXR*) increased more than 6.6-fold in August, while *GGPS* and *HMGR* increased 2.8- and 3.6-fold, respectively.

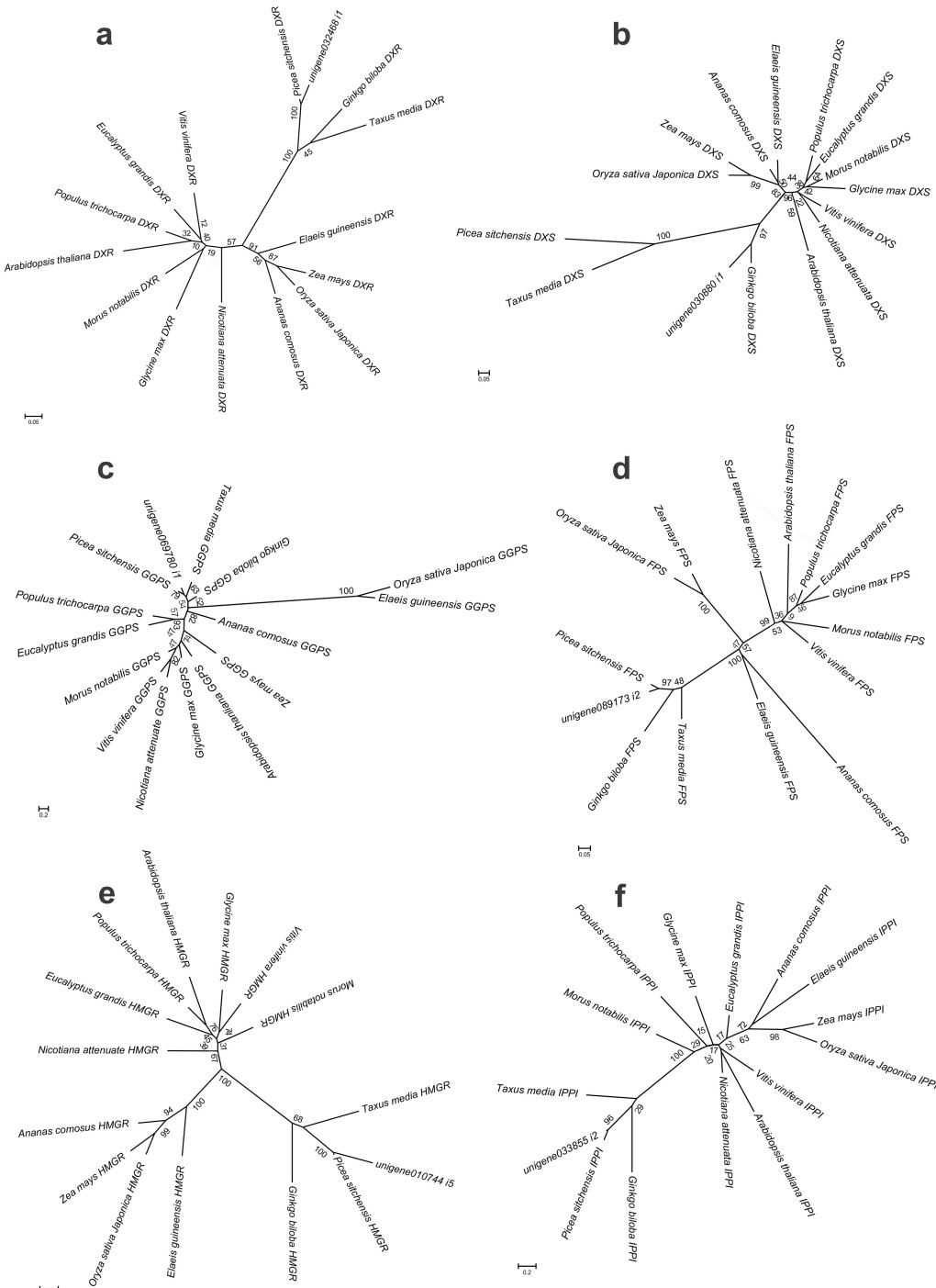

**Figure 9** **Bayesian phylogenetic analysis of the DXR, DXS, HMGR, IPPI, GGPS and FPS genes in 15 plant species.** They bootstrap values were given in present and the scale bar indicate 0.05, 0.05, 0.1, 0.2, 0.2 and 0.05 substitutions per site, respectively. (A) phylogenetic tree of DXR gene with JTT+I+G model, (B) phylogenetic tree of DXS gene with JTT+I+G model, (C) phylogenetic tree of GGPS gene with LG+G+F model, (D) phylogenetic tree of FPS gene with LG+I+G model, (E) phylogenetic tree of HMGR gene with JTT+G+F model, (F) phylogenetic tree of IPPI gene with JTT+I+G model.

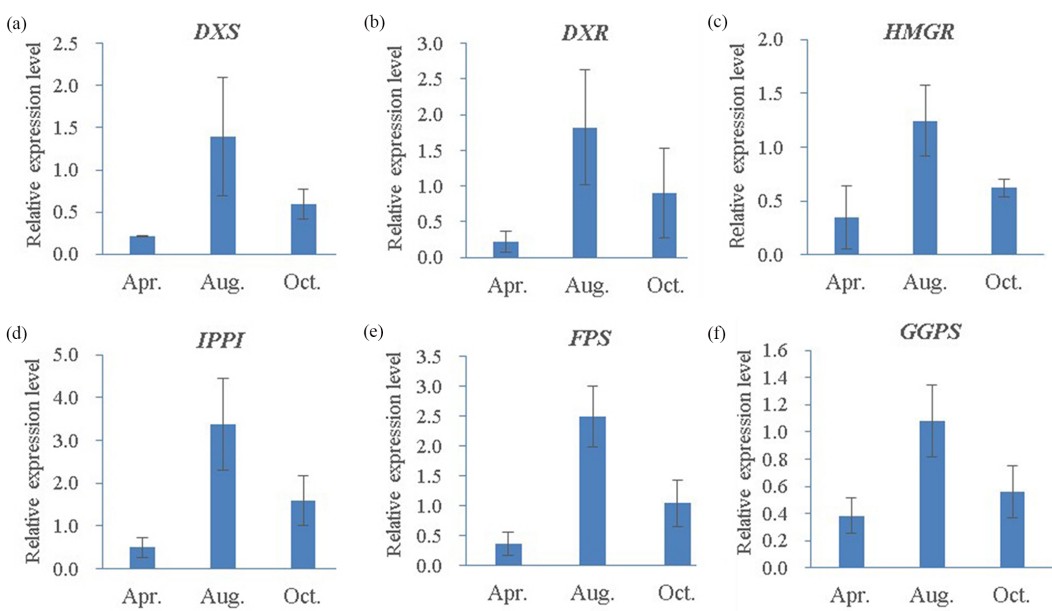

**Figure 10 Quantitative RT-PCR analysis of six candidate genes in three different oleoresin-yielding stages (Apr., Aug. and Oct.).** (A) DXS; (B) DXR; (C) HMRG; (D) IPPI; (E) FPS; (F) GGPS. Relative expression levels of qRT-PCR calculated using *Actin* as the internal control were shown in the $y$-axis. Error bars represent SD of the mean for three biological replicates.

Interestingly, compared to levels in October, all six genes increased ∼2-fold in August (Fig. 10). In summary, the six representative genes involved in terpenoid biosynthesis exhibited high expression levels during the high-oleoresin-yielding stage.

## DISCUSSION

Terpenoids play a critical role in the chemical and physical defense systems of conifers, allowing them to cope with attacks from herbivores and pathogens. In addition, as important secondary metabolites, terpenoids are widely used in industrial chemicals. Research on the mechanisms of terpenoid biosynthesis and increasing terpenoid yields has important economic value and biological significance. In this study, transcriptome analyses were performed on secondary xylem tissues of *Pinus taeda*. A total of 29,559,842 clean reads were obtained and assembled into 74,402 unisequences with a mean length of 1,459 bp. Only 31,586 unigenes (42.45%) were annotated by the five public databases. A total of 5,807 unigenes exhibited significant homology with sequences of *Picea sitchensis*. However, only 450 unigenes matched proteins from *Pinus taeda*. One possible explanation for this is the lack of available genomic information on *Pinus taeda* in public databases. Another possibility is related to the transcript material. Many studies on the genome of *Pinus taeda* are based on pine needles or seeds, and few studies have examined secondary xylem tissues (*Zimin et al., 2014*; *Westbrook et al., 2013*). After GO annotation, a mass of unigenes were identified as being involved in catalytic activity, binding, cellular processes, and metabolic processes. These results indicate that oleoresin biosynthesis in *Pinus taeda*

involves many unique processes and pathways. Furthermore, KOG functional classification and KEGG pathway analyses classified only 18,786 and 10,266 unigenes, respectively, far less than the number of unigenes annotated by the KOG (24,871) and KEGG (27,942) databases, as many genes in these databases have not been assigned functional classifications or pathways.

Based on the functional annotation results and previous research, a total of 372 unigenes were identified as involved in oleoresin biosynthesis. These unigenes were mainly speculated to encode terpenoid synthase, terpenoid backbone biosynthesis enzymes, cytochrome P450, ABC transporters, pathogenesis-related proteins, alcohol dehydrogenase, and aldehyde dehydrogenase, all of which have been previously reported as crucial for oleoresin biosynthesis. One previous study showed that ABC transporter expression is related to terpenoid production in loblolly pines (*Materna et al., 2006*). Pathogenesis-related proteins (PR5 and PR10) are commonly recognized as playing important roles in plant biological defense, particularly in the fight against fungal pathogens. PR5 proteins are induced by different phytopathogens in many plants and share significant sequence similarity with thaumatin-like protein (*Han et al., 2017*). The expression of PR10 proteins is also associated with many biotic stresses (*Hashimoto et al., 2004*; *EL-kereamy et al., 2009*). These results are consistent with previous reports that transcript levels of a *PR10* gene from *Prunus domestica* L. were significantly increased after two varieties of fungal infections (*Lois, Gallego & Campos, 2000*). Aldehyde dehydrogenase and alcohol dehydrogenase were reported to participate in the oxidation process of dihydroartemisinic aldehyde to dihydroartemisinic acid (*Teoh, Polichuk & Reed, 2009*) and artemisinic alcohol to artemisinic aldehyde, respectively (*Polichuk et al., 2010*). Cytochrome P450 family proteins are one of the largest classes of proteins involved in plant terpenoid secondary metabolites. In our study, a total of 67 unigenes of CYPs were identified and classified into 22 CYP subfamilies based on homology analyses. Among them were many important CYP subfamilies that participate in oxidation of various terpenoids, such as the CYP720B subfamily which is involved in the two consecutive oxidations processes in the formation of a series of diterpene resin acids with diterpene alcohols and aldehydes as substrates (*Hamberger et al., 2011*; *Geisler et al., 2016*). However, in some cases the third oxidation step of diterpene resin acids may be catalyzed by an aldehyde dehydrogenase (*Lupien et al., 1999*). In addition, many other subfamilies including CYP71A (*Bertea et al., 2001*), CYP71B (*Ginglinger et al., 2013*), CYP71D (*Schalk & Croteau, 2000*), CYP716A (*Tamura et al., 2017*), CYP701 (*Itkin et al., 2016*), CYP88 (*Helliwell et al., 2001*), CYP76B, and CYP76C (*Höfer et al., 2014*) have all been reported to be involved in the monooxygenation process of various terpenoids.

To date, many genes involved in the terpenoid biosynthetic pathway have been identified and well-studied in many other plants. In our study, 74 unigenes were identified to participate in terpenoid backbone biosynthesis. We further analyzed six representative unigenes (*DXR*, *DXS*, *HMGR*, *IPPI*, *GGPS*, and *FPS*) previously reported to be involved in the terpenoid biosynthetic pathway. In a previous study, two types of DXS (type I and type II) were identified to catalyze the first reaction of the MEP pathway and play a rate-limiting role in the production of MEP-derived isoprenoids (*Schmidt & Gershenzon,*

*2008*). In addition to DXS, DXR (*Cordoba, Salmi & León, 2009*) and IPPI (*Zhao et al., 2016*; *Berthelot et al., 2012*) also have rate-limiting roles in isopentenyl diphosphate and dimethylallyl diphosphate synthesis. It is well-accepted that HMGR is a key enzyme of the MVA pathway, which catalyzes the conversion of 3-hydroxy-methylglutaryl-CoA to mevalonate (*Cao et al., 2010*). Previous studies have suggested that over-expression of *HMGR* genes significantly increased the accumulation of terpenoids (*Jayashree et al., 2018*). FPP and GGPP are the precursors for the biosynthesis of sesquiterpenes and diterpenes, respectively. FPS is a key enzyme in isoprenoid biosynthesis, and is associated with plant growth and development (*Wang et al., 2018*). In *Pinus massoniana*, the expression levels of *GGPS* presented a substantially linear distribution when plotted against their corresponding oleoresin yields (*Chen et al., 2018*).

To further examine the evolutionary relationships of these six genes in plants, molecular phylogenetic trees were conducted based on their protein sequences, separately. The results showed that *DXS*, *DXR*, *HMGR*, and *IPPI* have the same unrooted topology, and the 15 amino acid sequences were divided into three distinct monophyletic groups. This indicates that these genes originate from gene duplication after species divergence (*Paszek & Górecki, 2016*; *Baldauf, 2003*; *Krushkal et al., 2003*; *Kim et al., 2009*). However, in the dicotyledon clade, the *DXS* and *IPPI* genes of *Arabidopsis thaliana* branched apart from other dicotyledon sequences. This may result from incomplete lineage sorting (*Chan, Ranwez & Scornavacca, 2017*). The *FPS* and *GGPS* genes exhibited relatively anomalous unrooted trees. The four monocotyledon *FPS* genes were divided into different branches, and the *Zea mays GGPS* gene was clustered into the dicotyledon clade. This may partly be due to the numerous unique mutations that can arise from rapidly evolving sequences (*Degnan, 2013*). Multiple sequence alignment analyses and conserved motif prediction results showed that the GGPS and FPS protein sequences were less conserved compared to the other four protein sequences, and for the DXS and DXR protein sequences, multiple identical motifs were discovered in different locations. This may be because upstream genes are expected to face stronger selective constraints and be more pleiotropic (*Yang, Zhang & Song, 2009*; *Lu & Rausher, 2003*).

## CONCLUSION

Transcriptome analyses of secondary xylem from loblolly pine were performed using RNA-Seq technology. A total of 74,402 unigenes were assembled from 29,559,842 high-quality reads, and 31,586 unigene sequences were annotated using five public protein databases. Based on the results of functional annotation and previous studies, 372 unigenes of terpenoid biosynthesis-related enzymes were identified. From these, six representative genes (*DXR*, *DXS*, *HMGR*, *IPPI*, *GGPS*, and *FPS*) were selected for multiple sequence alignment analyses, conserved motif prediction, phylogenetic analyses, and expression profile analyses. The results showed that all six genes exhibited high expression levels during the high-oleoresin-yielding phase. The protein sequences of DXS and DXR were more conserved and pleiotropic than the other sequences. In addition, most of these genes originated from gene duplication after the species divergence. Taken together, our results

provide an extensive perspective of regulatory factors involved in oleoresin biosynthesis of the loblolly pine, as well as a portfolio of candidate genes for future study. Further research should focus on exploring the patterns of evolutionary rates among genes of the terpenoid biosynthetic pathway.

## ACKNOWLEDGEMENTS

We thank the Science Corporation of Gene for its assistance in origin data processing and related bioinformatics analysis, and also thank the Yingde Research Institute of Forestry in Guangdong

### Funding

This work was supported by the National Key R&D Program of China (2017YFD0600502-3) and "948" program of State Forestry Administration P.R. China (2014-4-72). The funders had no role in study design, data collection and analysis, decision to publish, or preparation of the manuscript.

### Grant Disclosures

The following grant information was disclosed by the authors:
National Key R&D Program of China: 2017YFD0600502-3.
"948" program of State Forestry Administration P.R. China: 2014-4-72.

### Competing Interests

Jiehu Chen is employed by Science Corporation of Gene

### Author Contributions

- Jipeng Mao conceived and designed the experiments, performed the experiments, analyzed the data, contributed reagents/materials/analysis tools, prepared figures and/or tables, authored or reviewed drafts of the paper, approved the final draft.
- Zidi He performed the experiments, analyzed the data, contributed reagents/materials/analysis tools, authored or reviewed drafts of the paper, approved the final draft.
- Jing Hao analyzed the data, prepared figures and/or tables, authored or reviewed drafts of the paper, approved the final draft.
- Tianyi Liu analyzed the data, contributed reagents/materials/analysis tools, authored or reviewed drafts of the paper, approved the final draft.
- Jiehu Chen analyzed the data, authored or reviewed drafts of the paper, approved the final draft.
- Shaowei Huang conceived and designed the experiments, authored or reviewed drafts of the paper, approved the final draft.

### Data Availability

National Center for Biotechnology Information (NCBI) database, accession number: PRJNA482703.

## Supplemental Information

Supplemental information for this article can be found online at http://dx.doi.org/10.7717/peerj.6124#supplemental-information.

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
