# Peer review of "Identification, expression, and phylogenetic analyses of terpenoid biosynthesis-related genes in secondary xylem of loblolly pine (Pinus taeda L.) based on transcriptome analyses"

_PeerJ, doi:10.7717/peerj.6124_

## Round 0.1 · original submission · Major Revisions

Besides other important concerns, our reviewers remark that your study is actually a transcriptome study rather than the genome study expected from the title. They also highlighted that the full genome of Pinus taeda has been sequenced, and that therefore the assignment of your unigenes by comparison with other species is sub-optimal. Please address their concerns.

Reviewer 1 ·

Basic reporting

The study presents a transcriptome analysis of secondary xylem of loblolly pine. However, the English language of the manuscript should be improved to ensure that your international audience can clearly understand.

Experimental design

I notice that the loblolly pine genome has been sequenced by Zimin et al. (2014).Why did the authors still adopt a de novo method for transcriptome analysis? Once when the gene sequences have been obtained, it is more interesting to analyze their relative expression levels in such a specific tissue.

Validity of the findings

Six key genes (i.e. DXR, DXS, HMGR, IPPI, GGPS and FPS) involved in terpenoid biosynthetic pathways were selected for multiple sequence alignments, conserved motif prediction and phylogenetic analysis. Why did the authors choose these genes? Are they completely assembled? Are there additional homologs in the genome?

Additional comments

1. The title of the manuscript should be changed based on the content presented.
2. The loblolly pine genome should be mentioned in the introduction part. Besides, the reason to sequence the secondary xylem should be introduced.

Reviewer 2 ·

Basic reporting

The content of the manuscript is clear. Comprehensive background have been provided. Tables and Figures are up to the mark. In some sentences grammatical errors are noticed.

Experimental design

Methods are described in sufficient details.

Validity of the findings

Enough informative data are generated. Discussion and conclusion are well described.

Reviewer 3 ·

Basic reporting

Based on transcriptome data from RNA-seq of secondary xylem of loblolly pine, the authors obtained many genes related to terpenoid biosynthesis, and performed multiple sequence alignment analysis, conserved motif prediction and phylogenetic analysis on six key genes involved in terpenoid biosynthetic pathways. Overall, the manuscript provides some useful information for studying the regulatory factors involved in the terpenoid biosynthesis of loblolly pine. However, the manuscript still has some main issues need to be addressed. Moreover, the authors should check the manuscript carefully, and many minor mistakes need be corrected.

Experimental design

no comment

Validity of the findings

A major concern is that the results lack experimental validation on the expression patterns of key genes involved in terpenoid biosynthesis in different oleoresin-yielding stages (e.g. from low to high) by qPCR. The authors only sequenced the samples from a high oleoresin-yielding stage. Thus, the qPCR validations are important for identifying the candidate key genes.

Additional comments

Title
1. The title “Genome-wide” should be deleted because no genome-wide screening was conducted within the whole genome of the loblolly pine (Pinus taeda) in the study.

Introduction

1. Line 37-38, the biotic stresses obviously include insect herbivory, fungal attacks. Thus, the sentences should be reworded.

2. The sampled tissues in the study were from secondary xylem of loblolly pine. How secondary xylem tissues are associated with location of resin canals or terpenoid biosynthesis? The authors should add the related description in the Introduction section to provide more justification for secondary xylem as the studied tissues.

Materials and methods

1. Based on the NCBI SRA accession number PRJNA482703 provided by authors, I searched and found the sampled tissues named as cambium according to BioSample information(https://www.ncbi.nlm.nih.gov/biosample/9709534), which is not consistent with the RNA-seq samples in this manuscript (secondary xylem tissues). Cambium or secondary xylem? Please explain.

2. The biological replicates of RNA-seq should be stated clearly in the Method section.

3. Lines 70-73. The authors should provide more detailed information on sampling, such as the pictures of sampled tissue, sampled location in the trunk, and methods of sampling, which can be easily understood by readers.

4. Line 90. Illumina HiSeq 2000 or 4000?The inconsistence is shown between this manuscript and NCBI sequencing information (SRA experiments) (https://www.ncbi.nlm.nih.gov/sra?linkname=bioproject_sra_all&from_uid=482703 ).

5. Please explain why not use the genome of Pinus taeda as the transcriptome reference. After all, the genome of Pinus taeda has been sequenced and reported in 2014 (Zimin et al; Neale et al) and 2017 (Zimin et al).

Zimin A, Stevens K A, Crepeau M W, et al. Sequencing and assembly of the 22-Gb loblolly pine genome. Genetics, 2014, 196(3): 875-890.

Neale D B, Wegrzyn J L, Stevens K A, et al. Decoding the massive genome of loblolly pine using haploid DNA and novel assembly strategies. Genome biology, 2014, 15(3): R59.

Zimin A V, Stevens K A, Crepeau M W, et al. An improved assembly of the loblolly pine mega-genome using long-read single-molecule sequencing. Gigascience, 2017, 6(1): 1-4.


Results
1. Lines 130-132. The scientific name of species should be italic. Please carefully check this problem throughout this manuscript.

2. Lines 133-157. The descriptions about GO, KOG and KEGG annotations need to be revised because much annotation information is not important. For example, the largest functional group in each database was extensively highlighted by authors. The authors should conclude the results on the important functional classifications close to the topic of this manuscript.

3. Lines 158-168, line 199. To better understand the genes involved in terpenoid biosynthetic pathway, the authors should provide the schematic diagram of terpeniod biosynthetic pathway and label the corresponding key genes, such as some upstream or downstream genes (line 199).

4. Lines 199, 207,209, 211, …. If an abbreviation of gene name represents a specific gene, this gene name should be italic. Please check them throughout this manuscript.

5. Line 216. As shown in Fig. 7, the“orthologous to the Ginkgo biloba” is “orthologous to Elaeis guineensis”?


Discussion

Mirror errors, for example, line 228 “significanc”; line 236 “transript material”; line 290 “has”, and etc.

Conclusion
Lines 310-311. The authors claimed that the expression profiles analysis were performed in this study, however, there is no any expression profile results throughout this manuscript. Please revise this conclusion.

Reference
Some format errors are displayed among these references. For example, the abbreviation or full name of journals should be unified. Please check them carefully.

---

## Round 0.2 · accepted · Accept

Thank you for carefully addressing all of the issues highlighted by our reviewers.

# Reviewer 1 ·

Basic reporting

The authors have carefully modified and responsed my questions.

Experimental design

No additional comment.

Validity of the findings

No additional comment.

Additional comments

I suggest the title to be "Transcriptome identification, phylogenetic, and expression analyses of terpenoid biosynthesis-related genes in secondary xylem of loblolly pine (Pinus taeda L.)".

Reviewer 3 ·

Basic reporting

no comment

Experimental design

no comment

Validity of the findings

no comment

Additional comments

Thanks for responding to my comments. All my questions have been addressed.